# Nanoscale real-time detection of quantum vortices at millikelvin temperatures

A. Guthrie[1✉], S. Kafanov [1✉], M. T. Noble[1], Yu. A. Pashkin[1], G. R. Pickett [1], V. Tsepelin [1], A. A. Dorofeev[2,3], V. A. Krupenin[2,3] & D. E. Presnov [2,3,4]

Since we still lack a theory of classical turbulence, attention has focused on the conceptually simpler turbulence in quantum fluids. Reaching a better understanding of the quantum case may provide additional insight into the classical counterpart. That said, we have hitherto lacked detectors capable of the real-time, non-invasive probing of the wide range of length scales involved in quantum turbulence. Here we demonstrate the real-time detection of quantum vortices by a nanoscale resonant beam in superfluid $^4$He at 10 mK. Essentially, we trap a single vortex along the length of a nanobeam and observe the transitions as a vortex is either trapped or released, detected through the shift in the beam resonant frequency. By exciting a tuning fork, we control the ambient vortex density and follow its influence on the vortex capture and release rates demonstrating that these devices are capable of probing turbulence on the micron scale.

[1] Department of Physics, Lancaster University, Lancaster, UK. [2] Quantum Technology Centre, Moscow State University, Moscow, Russia. [3] Faculty of Physics, Moscow State University, Moscow, Russia. [4] Institute of Nuclear Physics, Moscow State University, Moscow, Russia. ✉email: andrew.guthrie@aalto.fi; sergey.kafanov@lancaster.ac.uk

Despite the long history of classical turbulence, and its very significant impact on human affairs we still lack a theory, since the governing Navier-Stokes equations are only soluble in quite specific simple situations. The importance of a general solution to these equations is underlined by its inclusion in the Clay Institute, Millennium Problems[1]. Quantum turbulence, the turbulence which occurs in quantum fluids, is much simpler than its classical counterpart, comprising single-quantised identical vortices. It can be thought of as an "atomic theory" of turbulence. This simplicity has led to the hope that understanding quantum turbulence can lead to insights into understanding the classical equivalent. Unfortunately, quantum turbulence is a microscopic phenomenon with significant length scales in the nanometre to micrometre range and we largely lack the tools to study liquid motions on these scales. Since quantum turbulence appears to behave as an ensemble of independent quantum vortices on short length scales but behaves very similarly to classical turbulence on larger scales, a critical scale length here seems to be the typical vortex separation. Thus, to advance our understanding, we need to be able to detect turbulence at the single-vortex level and on the scale of the vortex separation. Systems for trapping single vortices have been used previously[2–4], but with much larger dimensions. Using another approach, the SHREK project[5] has compensated for relatively large detector sizes by expanding the experimental space thereby making the scale of the flow patterns much larger.

Here we demonstrate the detection of single quantum vortices in superfluid $^4$He in real time. Vortices can be trapped on nanobeam resonators, signalled by an increase in the beam resonant frequency. This allows us to follow the capture of a single vortex, its presence in the trapped state and its subsequent release via reconnection with a nearby vortex in the surrounding vortex tangle, controlled by the excitation of a nearby quartz tuning fork transducer, advancing our capability to probe vortex tangles on much smaller length scales and faster timescales than has hitherto been possible.

## Results

The nanobeams we use for detecting single vortex events in real time have characteristic dimension <1 μm and response times faster than 1 ms. Such devices have recently emerged as highly sensitive probes of hydrodynamic[6,7] and ballistic $^4$He [8,9].

The beam used here has a vacuum frequency of 2.166 MHz, and at the measurement field of 5T has a $Q \sim 2.8 \times 10^3$. The beam is driven at a velocity of only a few millimetres per second. This is orders of magnitude below the expected velocity for the onset of turbulence production[10]. Therefore, in all our measurements the beam response is linear. The frequency width arises almost entirely from intrinsic losses in the beam material. Although while operating the beam in the liquid, we do see a vanishingly small increase in damping above the vacuum value arising from a very small level of acoustic emission into the liquid, but the contribution from the superfluid is essentially negligible.

Figure 1 displays schematically the measurement setup used for the single-vortex detection. Shown in the lower part of the figure, the doubly clamped, 70-μm-long, Al−Si$_3$N$_4$ nanobeam with a $130 \times 200$-nm cross-section provides the vortex detector.

Figure 2 shows the time evolution of the response of the nanobeam at each excitation frequency as an example of real-time interactions of the nanobeam with quantum turbulence. The time trace spans two similar consecutive interaction events. The trace clearly illustrates that the resonant frequency of the beam shifts significantly (by approximately five widths of the resonance) over a short period of time. We monitor such changes in real-time by the use of a 42-frequency comb

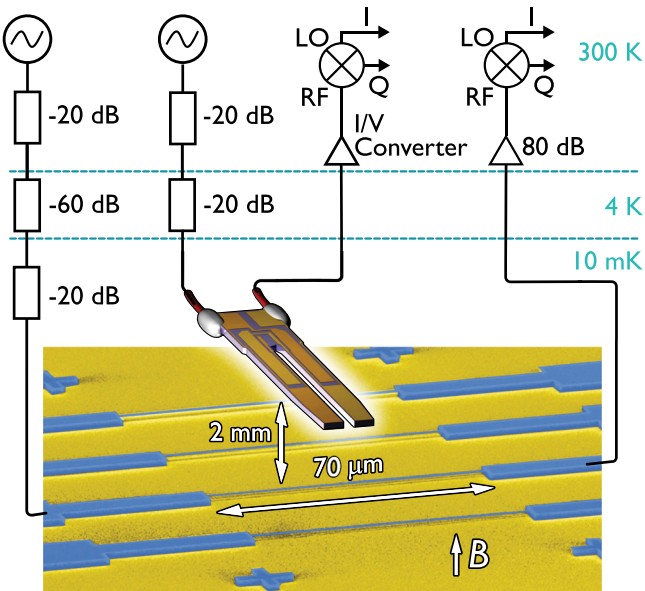

**Fig. 1 Schematic of the experimental setup.** A tuning fork generates quantum turbulence, whilst a 70-μm-long nanomechanical beam, suspended 1 μm above the substrate, acts as the detector. The beam and fork are driven by vector network analysers or signal generators through several stages of attenuation at various temperatures. The beam and fork signals are amplified at room temperature by an 80-dB amplifier and an I/V converter[22]. For a detailed description see the Supplementary Information.

produced by a multi-frequency lock-in amplifier[11,12]. The 2-ms time-analysis interval represents an optimal compromise between fast detection and the frequency resolution of the high-Q resonator.

The pattern of the events in the figure, with the frequency intermittently jumping from a low to a higher value and back again, is maintained over the many hundreds of such interactions we have recorded. Initially the beam frequency is low and stable. At time $\alpha$ (see figure) it gradually increases and stabilises in the region $\beta$ to $\gamma$ before abruptly resetting to the initial low-frequency state at time $\delta$. We can identify and associate each change with the successive stages of the nanobeam's interaction with the vortex tangle.

Referring to Fig. 2, the default state of the beam is that with the lowest frequency ($\delta_i\alpha_{i+1}$). This is the only beam response in turbulence-free superfluid, and we identify it with the vortex-free beam. In this state, the beam resonance frequency is reduced by 50 kHz from its vacuum value, consistent with the added effective mass contributed by the volume of superfluid displaced.

The damping of the beam in this state, inferred from the resonance width, is identical with that in vacuum. Therefore there is no significant added dissipation mechanisms in the presence of the superfluid, as expected from the low phonon and roton damping at a temperatures of ~10 mK[8].

We believe that the plateau state $\beta_i\gamma_i$, some ~3 kHz higher than the low state, represents the case where the nanobeam has trapped a singly quantised vortex along its entire length. The state is metastable, but will last for several days in the absence of local turbulence, and survives even if the beam motion is ceased, restarted or driven quite hard. However, upon restarting the turbulence source, the beam relaxes to the default state with the lower frequency described previously.

The identification of the capture of a singly quantised vortex by the beam is confirmed by several observations. First, the captive vortex generates an additional restoring force increasing

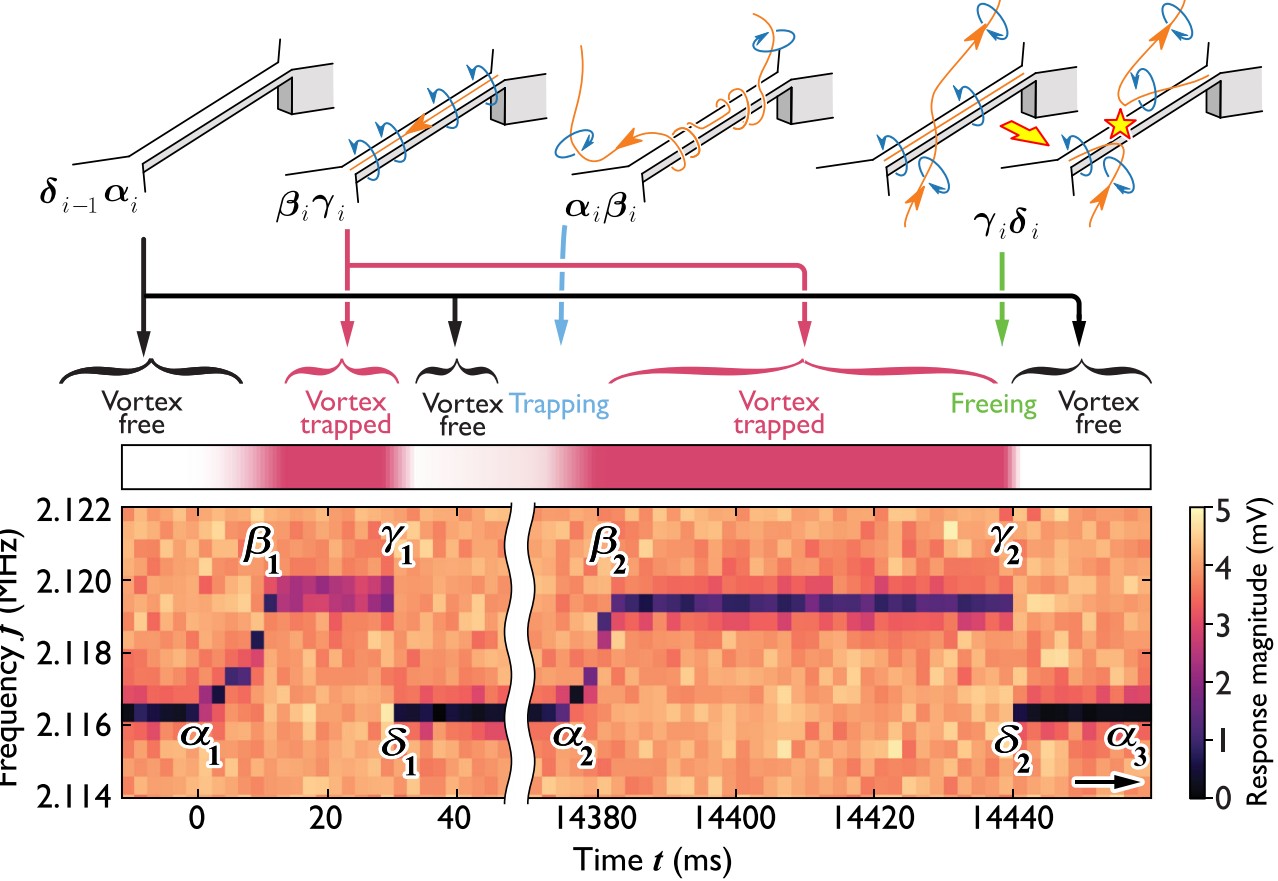

**Fig. 2 The magnitude of the nanobeam response at each excitation frequency against time taken from the start of the first event in heat-map format.**
Before point $\alpha_1$ the beam is in the default vortex-free state. Between $\alpha_1$ and $\beta_1$ a vortex interacting with the beam gradually raises the beam frequency by 3 kHz, finally becoming captured along the entire length of the beam at $\beta_1$. From $\beta_1$ to $\gamma_1$ the resonance is stable for 20 ms. The captured vortex interacts with a nearby vortex and at point $\gamma_1/\delta_1$ the system suddenly resets via reconnection of the trapped and attracted vortices and the beam resonance jumps back to the vortex-free state. After 14.35 s a second event at $\alpha_2$ occurs with similar features. The cartoons along the top of the figure sketch the broad processes involved, although the precise details of the capture and release mechanisms are not completely understood.

the beam's resonance frequency. This arises from the attractive interaction between the trapped vortex and its image in the nearby substrate. The interaction of the vortex with the parallel image vortex gives rise to a static force $\boldsymbol{F} = \rho \boldsymbol{v}_s \times \boldsymbol{\kappa}$, with $\rho$ the fluid density, $\boldsymbol{v}_s$ the superflow created by the image at the position of the beam's vortex and $\boldsymbol{\kappa}$ the circulation[2], which displaces the beam towards the substrate, thereby increasing its tension and thus yielding an increased beam resonant frequency when a vortex is trapped along it. For a fuller treatment see the Supplementary Materials.

At this point we should note that because the trapped vortex generates a circulation around the beam, there is a possibility that a Magnus force also comes into play, applying a vertical force as the beam moves transversely. If this force can distort the beam further, an additional frequency shift should occur. Any vertical Magnus force would necessarily be applied at the frequency of the resonant horizontal motion. Thus, for a Magnus force component to come into play, the vertical and horizontal resonant lines would have to overlap to excite a vertical motion. Bearing in mind that these are very narrow resonances ($Q \sim 3000$) and since the beam has a very non-symmetrical vertical/horizontal cross-section ($130 \times 200$ nm), and also very different terminal geometries in the vertical and horizontal planes, the horizontal and vertical resonances are likely to be very widely separated with a vanishing probability of any overlap. With our multi-frequency methods we can

readily probe for other resonances in the vicinity of the beam's horizontal frequency and have not seen anything. We can thus justify discounting this source of frequency change. It would be beneficial if we could indeed specifically excite such a Magnus mode as it might help to reduce the influence of the substrate.

Secondly, the damping of the beam hardly differs from that of the vortex-free or vacuum state, as expected, since the capture of a single vortex should not significantly change the acoustic emission[8], nor should it introduce any new dissipation mechanism, since the quasiparticle density is, in any case, essentially zero. Thirdly, the frequency of the upper plateau is almost always the same (3 kHz above the default state), supporting the idea of the capture of a singly quantised vortex. While double or even higher-order quantisation is not energetically unfavourable, it is hard to imagine any creation mechanism. Trapped multiply quantised vortices would yield discrete higher-frequency plateaus which have not been observed.

We now can attribute the transitions $\alpha_i\beta_i$ and $\gamma_i\delta_i$ between the default and metastable states to the capture and the release of a vortex by the beam. The latter process is always instantaneous on the scale of our detection time and is governed in some way by reconnection of the trapped vortex with a crossing vortex in the surrounding superfluid.

We have better understanding of the trapping process. A length of nearby vortex, lying more or less parallel to the beam, interacting with its negative image in the beam, will experience

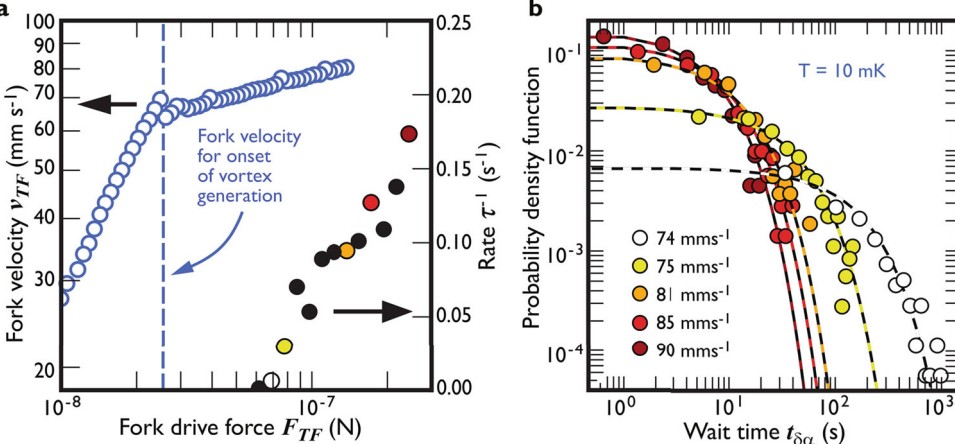

**Fig. 3 The capture process. a** The tuning fork velocity as a function of the applied force on the left axis and the rate of detected events by the beam on the right axis. The blue circles correspond to the tuning fork force-velocity dependence, while the symbols on the right show the beam detection rate at various fork forces. The dotted blue line corresponds to the onset of turbulence production by the tuning fork. **b** A probability density function of the wait time between events $t_{\delta\alpha}$ at the same fork velocities. The solid lines correspond to exponential fits, of the form $\propto \exp(-t/\tau)$. Note symbol colour matching between panels **a** and **b**, black points in panel **a** represent data from other runs not used for panel **b**. For details, see text.

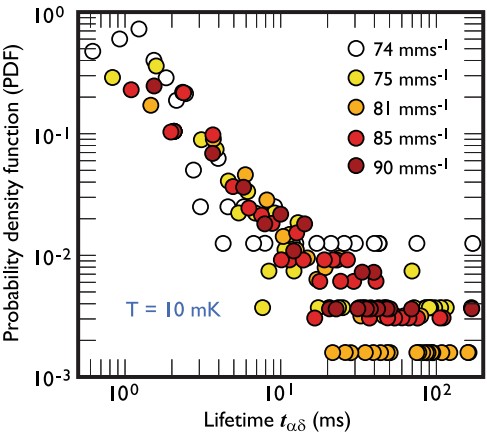

**Fig. 4 The release process.** A probability density function (PDF) of captured vortex lifetimes $t_{\alpha\delta}$ at selected fork velocities. The discrete data at long lifetimes are the result of single observed events. The data point colours reflect the same data as in Fig. 3.

both an attractive force towards the beam and a perpendicular transverse force parallel to the beam surface. Therefore, the vortex will gradually spiral into the beam, finally being trapped, as discussed by Griffiths[3], and as shown in the cartoon of Fig. 2.

The observed process of vortex capture by the beam is indeed a much more gradual process than the release. In addition to the "completed" events shown in Fig. 2, we also observe many embryonic cases which never fully develop and rapidly revert to the default state, implying that the vortex concerned does not achieve the fully trapping state, either through some failure of the process, or by premature dislodgement by reconnection with a second vortex.

We should emphasise here that the behaviour of the capture and release processes is completely different. We can show that by looking at the effect of the local vortex density on these two processes.

## Discussion

To discuss the results, we begin with the effect on the capture process shown in Fig. 3. The ambient vortex density is controlled by the velocity of the tuning fork[13]. At our temperatures, with

vanishing normal fluid density, and thus slow vortex decay, the action of the fork fills the entire experimental volume with essentially uniform vorticity. In panel **a** of the figure, we show the tuning fork velocity as a function of the driving force. The clear jump in the slope of the tuning fork response (marking greatly increased dissipation) indicates the onset of turbulence production, see for example references[14,15]. As discussed below, the results suggest that the trapping process is governed by the ambient vortex density in the immediate surrounding of the beam, say within a distance of a few micrometres.

In panel **a** we also plot a summary of the single-frequency measurements of the detection event rate, $\tau^{-1}$, defined as the inverse mean waiting time $\tau$ for an event to occur. The event rate increases with the fork's velocity confirming that the nanobeam probes the surrounding tangle density. We only detect vortices at tangle densities corresponding to fork velocities above 73 mm s$^{-1}$. At this velocity, the rate of detection is very low with the shortest waiting time between the events being ~40 s and the longest ~1000 s. (The gap in the fork driving force between the onset of vortex generation and actual detection by the beam, arises because at fork velocities just above onset, only a dilute gas of individual microscopic vortex rings is emitted. Only at higher velocities does the generated vortex ring density become high enough to favour the ring–ring collisions and reconnections necessary for the production of developed turbulence, see for example, Bradley et al.[15].

Panel **b** of Fig. 3 presents the probability density function (PDF) of the wait time, $t_{\delta\alpha}$, at five tuning fork velocities showing that the waiting time decreases with increased fork velocity, i.e., greater tangle density. The solid lines in the figure correspond to exponential distributions, of the form $\propto \exp(-t_{\delta\alpha}/\tau)$. Since it is known that turbulent tangles emit vortex loops following a similar exponential dependence[16,17], it appears that the capture process may well be governed by the stream of loops (especially the larger loops) emitted by the local tangle. However, whatever the detailed process, it is worth emphasising again that the capture process is governed by the surrounding vortex tangle density.

Once the vortex is captured, its lifetime follows a very different dependence. The release must depend on the proximity of another vortex for annihilation and thus should also carry information on the surrounding tangle. Figure 4 shows the probability density function of the measured lifetimes $t_{\alpha\gamma}$ of vortices on the nanobeam at five tuning fork velocities. First, the typical lifetime of a captured

vortex state is three orders of magnitude shorter than the wait time between events. Secondly, the data display no discernible dependence on the tuning fork velocity, showing that the release is insensitive to the overall vortex tangle density. This is surprising since we know that the captured state can exist essentially indefinitely if the vorticity is turned off (carefully to avoid dislodging the vortex in the process). Thus, although we understand the "on" and "off" states of the beam, we do not yet fully understand the processes leading to jumps between them.

Since, in the absence of ambient vorticity, the lifetime of the captured vortex is essentially infinite, the release process must be a result of interaction between the captured and external vortices. Although the PDF data of the captive lifetime shown in Fig. 4 is too scattered to indicate its functional form, we can use our range of lifetimes to make some rough estimates of the length scales involved. Optical measurements in superfluid helium[18,19] and simulations of quantum vortex behaviour[20] show that the time-scale, $t$, for vortex–vortex interactions displays a square root relationship with the vortex spacing $\delta$ as $\delta = A\sqrt{\kappa\, t_{\alpha\gamma}}$, where $\kappa$ is the circulation quantum and $A$ a constant of order 1, depending on the geometry of the approaching vortices[19]. This expression and our range of lifetimes of 3–100 ms (as in Fig. 4), suggests an initial vortex separation of 70–230 μm, consistent with both typical vortex tangle densities, and the distances reported by the optical measurements.

It is an interesting point, that all these observations concern the very low temperature regime where mutual friction is essentially absent. The approach to equilibrium in a vortex tangle under these conditions is conventionally attributed to Kelvin wave cascades on the individual vortices sourced by reconnections. How these ideas might be applied to the trapping and untrapping processes described here is not clear to the authors, but something, as we point out below, which may well be pursued with this experimental setup.

In conclusion, we demonstrate that nanobeams can be used as sensitive detectors of single vortex events, tracking their capture, interaction, and release with millisecond resolution, thereby able to probe the local vortex line density. We foresee that we could readily manufacture multiplexed arrays of such beams with the ability to probe the spatial and temporal evolution of a complex vortex tangle again with millisecond and potentially single-vortex resolution. Looking further ahead, by capturing a single-vortex in an engineered trapping configuration, we may well be able to study the dynamics of Kelvin waves on the captive vortex, a much anticipated goal in quantum turbulence research[21].

## Methods

**Device description**. The nano-electromechanical device system (NEMS) consists of a doubly-clamped aluminium-on-silicon nitride (Al−on−Si$_3$N$_4$) composite nanobeam. The beam's dimensions are defined lithographically, with length $l = 70$ μm and width $w = 200$ nm. The 100-nm-thick Si$_3$N$_4$ layer determines the mechanical properties of the beam, while the Al layer allows the magnetomotive excitation and measurement of the beam motion. The combined thickness of the aluminium and silicon nitride layers is $t = 130$ nm, with a combined density of 3062 kg m$^{-3}$. The vacuum frequency of the fundamental mode is determined experimentally to be $f_0 = 2.166$ MHz. The nanobeam is suspended approximately $d \sim 1$ μm above the silicon substrate. The experiment is housed in a brass experimental cell containing superfluid $^4$He at a temperature of 10 mK, mounted on the mixing chamber of a cryogen-free dilution refrigerator.

**Measurement scheme**. The nanobeam response is probed by a magnetomotive detection scheme. The Lorentz force driving the nanobeam is generated by an AC current passed through the nanobeam in a perpendicular magnetic field supplied by a large external superconducting solenoid. The beam motion in the magnetic field generates a Faraday voltage which is detected as a drop in the transmitted signal. For characterisation of the nanobeam, a vector network analyser is used both to supply the AC current and to acquire the transmitted response measured as a function of frequency. The resulting Lorentzian resonance curve is fitted to obtain the nanobeam velocity, $v$, and force, $F$, using previously established methods[6].

To perform time-dependent resonance tracking, two phase-sensitive lock-in measurement techniques are employed: single-frequency detection, and multi-frequency detection. For single-frequency detection we use a signal generator supplying a fixed-frequency constant AC signal to the nanobeam input, with the nanobeam output connected to a high-frequency (SR844) lock-in amplifier. With the driving frequency fixed on resonance, any change in the nanobeam resonance frequency is detected as a drop in the measured signal.

For simultaneous detection at multiple frequencies, a multi-frequency lock-in amplifier (MLA)[11] is used in place of the signal generator and high-frequency lock-in. The MLA instrument employs a frequency comb comprising integer multiples $n_i$ of a base tone $f_b$ such that all measurement frequencies $f_i$ satisfy $f_i = n_i f_b$. For distinguishing between tones, the measurement time $t_m$ must be larger then the inverse separation between frequencies $t_m > 1/f_b$. This constrains the time resolution and frequency spacing of the instrument, and faster measurements require the frequencies to be spaced further apart. It is also pertinent to note that the non-linearities of the resonator will cause mixing between the frequency tones although the use of low-excitation drives avoids this problem[12].

For both resonance tracking techniques, an oscilloscope is used in conjunction with the lock-in demodulation in order to record vortex capture events. The lock-in demodulated signal at the vortex-free resonant frequency of the beam is monitored by the oscilloscope, which triggers the lock-in amplifier to record data when the signal strength falls sufficiently from the resonance frequency shift. For single-frequency measurements the fall and subsequent rise in the signal would then yield the event lifetime. In the multi-frequency measurements the recorded data is fitted with a Lorentzian peak to obtain the beam's resonate frequency as a function of time, and the lifetime is then found from this data.

Similarly, the tuning fork is measured with a vector network analyser, using an I–V converter[22] (trans-impedance amplifier) to recover the signal which can then be used to find the fork velocity[23,24]. The driving force on the fork can be found from the drive signal using well-established techniques[23,24].

## Data availability

All the data contributing to the study described in this paper, including descriptions of the data sets, are available from Lancaster University's data repository at https://doi.org/10.17635/lancaster/researchdata/424.

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

## Acknowledgements

We thank all members of Lancaster University ULT group as well as A.I. Golov, O. Kolosov, P.V.E. McClintock and W.F. Vinen for helpful discussions. This research was supported by UK EPSRC grant no. EP/P022197/1 and the EU H2020 European Microkelvin Platform (grant agreement 824109). The MSU team was supported by the Interdisciplinary Scientific and Educational School of Moscow State University "Photonic and Quantum Technologies. Digital Medicine", the research infrastructure of the "Educational and Methodical Center of Lithography and Microscopy", M.V. Lomonosov Moscow State University was used.

## Author contributions

The nanomechanical samples were fabricated by A.A.D., V.A.K. and D.E.P. The idea of the experiment was formulated by A.G., S.K., M.T.N., Y.A.P. and V.T. and performed by A.G., S.K. and M.T.N. The data analysis was made by A.G. and M.T.N. The interpretation of the results was performed by A.G., S.K., M.T.N., Y.A.P., G.R.P. and V.T. and the manuscript was mainly written by A.G., M.T.N., G.R.P. and V.T.

## Competing interests

The authors declare no competing interests
