## [Peer Review File · Nature Communications]

REVIEWER COMMENTS

Reviewer #1 (Remarks to the Author):

The manuscript by Guthrie et al describes measurements of a frequency shift of a nanobeam oscillating in superfluid 4He at very low temperature in the presence of quantum turbulence generated by a quartz tuning fork. Repeatable shifts of the frequency up and down are observed and interpreted as a capture/release events of a single quantized vortex. This observation suggests that a nanobeam device can be used as a detector of quantum turbulence with unprecedented temporal and spatial resolution, which opens new horizons in studies of quantum turbulence. I find this result fascinating and appropriate for publication in Nature Communications. Before I can recommend the publication, I ask the authors to consider the suggestions below to improve the presentation and to make it more suitable for the Nature Communications readership.

1. The authors use an old argument of potential similarities between classical and quantum turbulence as their main motivation. In recent years, however, understanding emerged that one of the most remarkable aspects of quantum turbulence is the coexistence and interplay between classical features at the scales larger than the intervortex distance and quantum features at smaller length scales. From the size of the used device I conclude that it works on the quantum side. This is an excellent achievement by itself, but I do not see how such sensor could potentially contribute to understanding of classical turbulence. The authors should clarify their motivation.
2. Essentially no account for the previous works is given. Observation of a single vortex by trapping it along a thin oscillating object is certainly not new (the new features are accessible length and time scales). Earlier works by W.F. Vinen and R. Zieve with colleagues should be acknowledged. On the classical side of the quantum turbulence range of scales, it is worth mentioning sensor miniaturization work within the SHREK project (miniature Pitot tubes and hot nanowires). It should also be explained why seemingly quantitative improvement in spatial and temporal resolution is actually a jump to unexplored territory (by comparing to the relevant quantum turbulence scales).
3. The manuscript contains certain strong conclusions which I find not fully backed up by the presented data:
 - a) The claim that frequency up processes measure "the local vortex tangle density" (page 6). I see that the PDF of those processes depends on the fork drive, which supposedly affects the overall vortex density. I do not see any proof of the "locality" of sensing (which could be provided for example with the data from two devices at different distances from the fork or using one device and

introducing some obstacle between the vortex generator and the sensor). Also it is not clear what length scale the authors consider "local".

b) The claim of "excellent agreement" (page 7) between time-distance dependence during a vortex reconnection (from earlier experimental and numerical works) and the present observation of the frequency down events. In the works the authors refer to, a reconnection between two freely moving vortices far from the container walls was studied. In the author's own interpretation, one of the vortices in this work is trapped and immobile. Additionally, solid surfaces which strongly affect the flow are present in the immediate vicinity. How does this affect dynamics remains open without further (possibly numerical) investigations. So it is unclear how such comparison with vortex reconnection dynamics studied earlier can be used as an argument in favor of the author's interpretation of the frequency down events.

c) The claim that trapping of a vortex should not "introduce any new dissipation mechanism" (page 5). In fact, circulating flow field does change the scattering of the quasiparticles (that's an origin of the mutual friction). It is simply in the settings of the present experiment the density of bulk quasiparticles is low and the instrumental dissipation is high and thus such effect could not be observed.

The authors should reformulate the claims or provide more experimental/numerical evidence to back up their statements.

4. As an impact of this work the authors suggest a new sensor for quantum turbulence studies. An important characteristic of a sensor, besides its spatial and temporal resolution, is how invasive it is for the studied flow. This aspect is not mentioned now at all, but should be included in the discussion. In fact, it seems that presence of a nearby wall is essential for the demonstrated sensitivity (vortex image effect). But such wall would strongly affect also the studied turbulence. Boundary layers in quantum turbulence is an interesting subject but still only a limited subfield. The authors should present a broader view on potential applications.

Reviewer #2 (Remarks to the Author):

Quantum turbulence is currently not only an important branch of low temperature physics but also attracts interest from diverse fields such as classical fluid dynamics, nonlinear and non-equilibrium

science etc. Quantum turbulence has been studied for more than a half century, but the recent novel experiments and numerical simulations make innovation. The most important characteristic of quantum turbulence is that it consists of quantized vortices which are stable topological defects. The Lancaster experimental group of the authors of this article is the leading one in the world and has made a substantial contribution to the field. The major claim of this article is that the authors succeeded in observing the real-time detection of a quantized vortex by using a nanoscale resonant beam in superfluid ^4He at 10mK. A tuning fork supplies vortices, one of which is trapped by the nano beam. The trapped vortex survives along the beam with a life time, then leaves the beam through reconnection with the surrounding tangle. Almost all previous experiments addressed a group of quantized vortices, thus the story of a single quantized vortex is very novel and surprising. The work is convincing. The explanation of physics is clear and easy to understand. Before the acceptance, however, I would ask the authors to make a comment for the following point.

The experiments are performed at 10mK, where there is no mutual friction which works as the dissipative mechanism. I am just worried about this effect. The vortices do not decay by the mutual friction, which would be a good news. What are other effects? (1) A single vortex should be attracted to the nanobeam by the image force as discussed in the supplementary material. Some dissipative mechanism could carry the system into the equilibrium state, in other words capture the vortex properly. Is this possible even in the absence of mutual friction? (2) The release of a trapped vortex is caused by the reconnection with another surrounding vortex. The reconnections are encouraged by mutual friction too. Under the mutual friction, two vortices approach each other with making themselves antiparallel locally, but this mechanism lacks in the present situation. (3) Without mutual friction the trapped vortex may fluctuate and excite Kelvin waves along it.

Reviewer #3 (Remarks to the Author):

The manuscript reports development of a new probe for vortices in superfluid helium, a nanoscale beam which oscillates at a few MHz. The authors observe a frequency change when a vortex is trapped along the beam. They find that the vortex becomes trapped or free through different mechanisms, based on the manner in which the frequency transitions, how long it takes for events in each direction to occur, and the dependence on the surrounding vortex density.

This is all of interest to the helium community. The authors do not yet understand several aspects of the experiment, including crucially the mechanisms by which a vortex becomes trapped on the beam, is freed from the beam, and affects the beam's vibration frequency. Thus the manuscript is

more a proof-of-principle, with little by way of quantitative results, and may not have much appeal outside the immediate community.

The work would be strengthened with better models for the above processes. The authors' discussion for the mechanism of the frequency shift is particularly unconvincing. They first dismiss the Magnus force based on the experimental observation that the frequency change is nearly independent of the beam's velocity. However, that is what should happen: the Magnus force has magnitude proportional to velocity, but so does the momentum of the beam. The ratio of these two, which is velocity-independent, determines the frequency shift.

The supplemental material gives a different argument against the Magnus force, that its direction changes sign during each cycle. On general principle, a force that varies in sync with an oscillator is exactly what would be expected to affect the oscillation. It seems that the authors assume that the frequency shift originates in a change in the average position of the nanobeam, rather than considering the possibility that adding a vertical component to the mainly horizontal motion of the nanobeam might also shift the frequency of the mode.

The authors suggest another mechanism, that the interaction of a trapped vortex with the nearby surface creates a steady vertical force that changes the equilibrium location of the nanobeam and hence its tension. They refer to the force as "repulsive," but in fact it is attractive. (The image vortex is directed opposite to the physical vortex.) Perhaps the authors caught this error while editing; they do say the beam will "sag," which is consistent with an attractive force. The expected frequency shift from this mechanism is a few times smaller than what is observed. The Magnus force may need further consideration.

For the trapping of vortices from the nanobeam, there seems to be a gap between the velocity where the tuning fork first produces turbulence and that at which vortices appear on the nanobeam. Do the authors have any explanation? I also wonder about the schematic in Figure 2, with a vortex terminating midway along the beam. Is this physically reasonable? Where would such a vortex come from, since a finite segment certainly cannot terminate in the bulk fluid?

The authors note that the timing for the process of freeing a vortex strangely requires the presence of vortices in the fluid but does not depend on their density. However, from the lifetime of the vortex on the nanobeam they then calculate a typical vortex separation. How can this be meaningful if they know that changing the vortex separation does not alter the distribution of lifetimes? I also question the calculation. Why is 3 ms used as the lower end of the trapped lifetime? There are clearly many events in Figure 4 with lifetime 1 ms or even shorter. Secondly, the lifetime and

separation ranges seem inconsistent. Each extends about a factor of 3, but the square-root relationship suggests that a factor of 3 in distance should correspond to a factor of 10 in lifetime.

Can the probability distribution of Figure 4 further test ideas on trapping and release? Is the square root relation between lifetime and distance consistent with Figure 4? Or what about the idea that many incipient vortices are dislodged by the same mechanism that eventually frees the fully trapped vortices? The lifetime curves of Figure 4 overlay, but the curves of Figure 3b showing the time to trap do not. Doesn't this mean that the mechanism corresponding to Figure 4 cannot also be the main factor in Figure 3b?

I also have two minor questions for the authors. What are the black circles in Figure 3a? The other colors correspond to curves labelled in Figure 3b. Also, how are the PDFs of Figure 4 constructed? I do not understand why the lifetime values at small times do not agree better across the data sets, especially for the 75 mm/s and 90 mm/s sets which judging from the long-time region have an almost identical number of events.

Comments to reviewer 1

We were pleased to see that this reviewer found the result “*fascinating and appropriate for publication in Nature Communications*”.

Taking the reviewer’s points below in order:

1. *The authors use an old argument of potential similarities between classical and quantum turbulence as their main motivation. In recent years, however, understanding emerged that one of the most remarkable aspects of quantum turbulence is the coexistence and interplay between classical features at the scales larger than the intervortex distance and quantum features at smaller length scales. From the size of the used device I conclude that it works on the quantum side. This is an excellent achievement by itself, but I do not see how such sensor could potentially contribute to understanding of classical turbulence. The authors should clarify their motivation.*

We certainly believe what we wrote. A serious puzzle is that stated above by the reviewer, that at longer scales classical and quantum turbulence appear to behave similarly. That implies a measure of co-operative interaction between vortices on the larger scales to produce larger scale “eddies”. Of course, we are not expecting that quantum turbulence studies will solve the classical problem, but we hope they will at least provide insight.

We have rewritten the relevant section in the introduction to say just that and no more:

Since we still lack a theory of classical turbulence, attention has focused on the conceptually simpler turbulence in quantum fluids. If we can reach a better understanding of the quantum case, this may provide further insight into the classical counterpart.

2. (a) *Essentially no account for the previous works is given. Observation of a single vortex by trapping it along a thin oscillating object is certainly not new (the new features are accessible length and time scales). Earlier works by W.F. Vinen and R. Zieve with colleagues should be acknowledged. On the classical side of the quantum turbulence range of scales, it is worth mentioning sensor miniaturization work within the SHREK project (miniature Pitot tubes and hot nanowires).*

We should certainly acknowledge the earlier work of Vinen and Zieve *et al.*

We have added the reference. We have also added a note on the SHREK project:

While single vortex trapping systems have been used previously^{6–8}, but with much larger dimensions, the current measurements advance our capability to probe vortex tangles on much smaller scales and faster timescales than has hitherto been possible. Using another approach, the SHREK project⁹ has compensated for relatively large detector sizes by expanding the experimental space thereby making the scale of the flow patterns much larger.

[6] W. F. Vinen, Proc. R. Soc. Lon. Ser.-A 260, 218 (1960).

[7] D. J. Griffiths and J. F. Allen, Proc. R. Soc. Lon. Ser.-A 277, 214 (1964).

[8] R. J. Zieve, C. M. Frei, and D. L. Wolfson, Phys. Rev. B 86, 174504 (2012).

[9] B. Rousset, *et al.*, Rev. Sci. Instr. 85, 103908 (2014).

- (b) *It should also be explained why seemingly quantitative improvement in spatial and temporal resolution is actually a jump to unexplored territory (by comparing to the relevant quantum turbulence scales).*

We have added a piece emphasizing this:

Quantum turbulence appears to behave as an ensemble of independent quantum vortices on short length scales but behaves very similarly to classical turbulence on larger scales. A critical scale length here seems to be the typical vortex separation and the present device will allow us to probe behaviour on scales shorter than this, taking us into a completely new experimental regime.

3. *The manuscript contains certain strong conclusions which I find not fully backed up by the presented data:*

- (a) *The claim that frequency up processes measure “the local vortex tangle density” (page 6). I see that the PDF of those processes depends on the fork drive, which supposedly affects the overall vortex density. I do not see any proof of the “locality” of sensing (which could be provided for example with the data from two devices at different distances from the fork or using one device and introducing some obstacle between the vortex generator and the sensor). Also, it is not clear what length scale the authors consider “local”.*

Sorry, that confusion arises from our poor use of the word “local”. We really mean “ambient”. We have rewritten and expanded the relevant section to correct this and have added a note to suggest the distance over which we believe the wire probes:

We should emphasise here that the behaviour of the capture and release processes is completely different. We can show that by looking at the effect of the ambient vortex density on these two processes.

We begin with, the effect on the capture process shown in Fig. 3. The ambient vortex density is controlled by the velocity of the tuning fork¹³. At our temperatures, with vanishing normal fluid density, and thus slow vortex decay, the action of the fork fills the entire experimental volume with essentially uniform vorticity. In panel (a) of the figure, we show the tuning fork velocity as a function of the driving force. The clear jump in the slope of the tuning fork response (marking greatly increased dissipation) indicates the onset of turbulence production, see for example references^{16,17}. As discussed below, the results suggest that the trapping processes is governed by the ambient vortex density in the immediate surrounding of the beam, say within a distance of a few micrometres.

- (b) *The claim of “excellent agreement” (page 7) between time-distance dependence during a vortex reconnection (from earlier experimental and numerical works) and the present observation of the frequency down events. In the works the authors refer to, a reconnection between two freely moving vortices far from the container walls was studied. In the author’s own interpretation, one of the vortices in this work is trapped and immobile. Additionally, solid surfaces which strongly affect the flow are present in the immediate vicinity. How does this affect dynamics remains open without further (possibly numerical) investigations. So it is unclear how such comparison with vortex reconnection dynamics studied earlier can be used as an argument in favor of the author’s interpretation of the frequency down events.*

Yes, “excellent agreement” was not really what we were trying to express. These are all very new ideas and we wanted to indicate that what we see is not out of line with other related experiments. We have replaced “excellent agreement” with “consistent with” which is what we actually meant:

This expression and our range of lifetimes of 3 to 100mscommands (as in Fig.4), suggests an initial vortex separation of 70 to 230 μm , consistent with both typical vortex tangle densities, and the distances reported by the optical measurements.

- (c) *The claim that trapping of a vortex should not “introduce any new dissipation mechanism” (page 5). In fact, circulating flow field does change the scattering of the quasiparticles (that’s an origin of the mutual friction). It is simply in the settings of the present experiment the density of bulk quasiparticles is low and the instrumental dissipation is high and thus such effect could not be observed.*

We have added a note that this is governed by the vanishing quasiparticle density:

Secondly, the damping of the beam hardly differs from that of the vortex-free or vacuum state, as expected, since the capture of a single vortex should not significantly change the acoustic emission⁴, nor should it introduce any new dissipation mechanism, since the quasiparticle density is essentially zero. Thirdly, the frequency of the upper plateau is almost always the same (3kHz above the default state), supporting the idea of the capture of a singly-quantized vortex.

4. *As an impact of this work the authors suggest a new sensor for quantum turbulence studies. An important characteristic of a sensor, besides its spatial and temporal resolution, is how invasive it is for the studied flow. This aspect is not mentioned now at all, but should be included in the discussion. In fact, it seems that presence of a nearby wall is essential for the demonstrated sensitivity (vortex image effect). But such wall would strongly affect also the studied turbulence. Boundary layers in quantum turbulence is an interesting subject but still only a limited subfield. The authors should present a broader view on potential applications.*

These are preliminary measurements. The presence of the wall is not central to the operation of the system but in the first instance it is very much easier to create the nanostructures without departing too much from current fabrication techniques. Certainly, creating a beam, with the backing substrate removed over a large length of the beam allowing it to “free float” in a larger open volume, is quite feasible. There will still be a frequency shift from the end effects. However, we are taking one step at a time here.

Comments to reviewer 2

We were pleased to see that this reviewer noted that: “Almost all previous experiments addressed a group of quantized vortices, thus the story of a single quantized vortex is very novel and surprising. The work is convincing. The explanation of physics is clear and easy to understand. Before the acceptance, however, I would ask the authors to make a comment for the following . . .”

We were also very pleased to note that the reviewer 2 asks us to consider/discuss three points concerning the lack of mutual friction in the very low temperature regime in which we work, since the points raised are ones which have greatly exercised us for a very long time. We know that on small scales quantum turbulence behaves as an ensemble of single randomly distributed quantized vortices, but that on larger scales it mimics classical turbulence with large-scale features. In earlier experiments we have seen the evolution of these large-scale features from an initial gas of microscopic vortex loops (see Bradley *et al.*, PRL **95**, 035302 (2005)). Granted this work was in the context of superfluid ^3He but the same strictures apply. Large-scale features comprising singly-quantised vortices seem inevitably to imply a measure of cooperative interaction to form, for lack of a better word, bundles. As the reviewer points out, this presupposes that some sort of dissipation must operate to stabilize the final state and we seem to be lacking any candidate for the dissipation needed. In short, we are well aware of this problem, to the point of wondering whether there may be some critical generic element operating in both quantum and classical turbulence that we are somehow missing.

However, in the current context, this paper is largely on the new detector and its novel properties and potential and it is not easy to encapsulate the above ideas with the brevity needed here.

Nevertheless, they are important considerations and we have done our best to sketch in at least a minimum comment.

Taking the reviewer’s queries in detail:

1. *A single vortex should be attracted to the nanobeam by the image force as discussed in the supplementary material. Some dissipative mechanism could carry the system into the equilibrium state, in other words capture the vortex properly. Is this possible even in the absence of mutual friction?*

Since writing this paper we have reached a better understanding the trapping process, and now, following Griffiths’ calculation that an external vortex will spiral into the beam under the influence of its image. The process certainly progresses slowly compared with the release, which suggests that it is a “difficult” process. How equilibrium is reached in a “dissipation-less” zero-normal fluid regime is a problem for all quantum turbulence behaviour under these conditions. What we should probably expect is that a shower of Kelvin waves is launched along the pre-existing “incoming” vortex. We simply do not know at this stage. However, the trapping process does indeed happen. We need to think further about exactly how.

2. *The release of a trapped vortex is caused by the reconnection with another surrounding vortex. The reconnections are encouraged by mutual friction too. Under the mutual friction, two vortices approach each other with making themselves antiparallel locally, but this mechanism lacks in the present situation.*

Reconnections are perhaps the easiest part to grasp. The sharp kinks so created, provide a source of Kelvin waves which can then take part in a further wave cascade.

3. *Without mutual friction the trapped vortex may fluctuate and excite Kelvin waves along it.*

Indeed, this may well be the case, as we note above (despite the probable energy barrier implied when the core is trapped on the beam). We also allude to this in the very final sentence of the paper:

Looking further ahead, by capturing a single vortex in an engineered trapping configuration, we may well be able to study the dynamics of Kelvin waves on the captive vortex, a much-anticipated goal in quantum turbulence research²³.

Comments to reviewer 3

We were pleased to see that this reviewer found the result “*This is all of interest to the helium community.*” Happily, this has been a very useful review for us, since responding has clarified our thinking on a number of the issues raised.

Taking the reviewer’s points below in order:

1. *This is all of interest to the helium community. The authors do not yet understand several aspects of the experiment, including crucially the mechanisms by which a vortex becomes trapped on the beam, is freed from the beam, and affects the beam’s vibration frequency. Thus the manuscript is more a proof-of-principle, with little by way of quantitative results, and may not have much appeal outside the immediate community.*

As a result of the rethink precipitated by this reviewer’s comments we now have now formed a pretty clear picture of the capture process and we have also revised our understanding of the frequency shift (see below).

While we still do not understand the release process, we do believe that this work is a major step forward in the detection of vortices on a microscopic scale. While vortex trapping has been used previously, notably in the early days by Vinen, this is the first time it has been done on the microscopic scale and gives us a new tool in looking at vortex ensembles in the length-scale region associated with the cross-over from “classical” to quantum behaviour. Thus, we believe it certainly will be of interest.

2. *The work would be strengthened with better models for the above processes.*

As mentioned above, we now have a better grasp of the capture process. However, we should emphasize that the trapping and untrapping processes are utterly different. That was a surprise. The trapping process clearly faces obstacles in progressing. It not only proceeds gradually, with the frequency increasing over of order 10 ms (which we assume is the time it takes the trapping vortex to envelop the whole length of the beam), but the majority of “attempts” fail, with the process only partially succeeding and then a sudden dropping back to the untrapped state.

When a length of vortex approaches the beam two forces come into play, the attraction between the vortex and image, and the transverse force exerted on the vortex by the image vortex flow field. Initially we believed that the transverse force would throw off the incoming vortex and prevent any trapping. However, in the light of the calculation by Griffiths (Proc. R. Soc. Lon. Ser.-A **277**, 214 1964) we have revised our picture of the capture process which is now consistent with the measurements that the probability is a function of the ambient vortex density.

In contrast, the untrapping process is instantaneous on our experimental timescales and we believe requires the intervention of a further neighbouring vortex to initiate the process (since we know that clearing vortices from the system allows the trapped vortex to stay trapped essentially for infinite time). That said, as the referee points out, we find no correlation between the untrapping probability and the ambient vortex density.

We have further emphasized the difference between the trapping and untrapping processes at various points in the text. We have also retitled the captions for Fig.s 3 and 4 to highlight this difference.

FIG. 3. (Colour online) a. **The capture process.** The tuning fork velocity as a function of the applied force on the left axis and the rate of detected events by the beam on the right axis. ...

FIG. 4. (Colour online) **The release process.** A probability density function (PDF) of captured vortex lifetimes $t_{\alpha\beta}$ at selected fork velocities. ...

We have also changed the text introducing the trapping/untrapping processes as follows:

We now can attribute the transitions $\alpha_i\beta_i$ and $\gamma_i\delta_i$ between the default and metastable states to the capture and the release of a vortex by the beam. The latter process is always instantaneous on the scale of our detection time and is governed by reconnection of the trapped vortex with a crossing vortex in the surrounding superfluid.

We have better understanding of the trapping process. A length of nearby vortex, lying more or less parallel to the beam, interacting with its negative image in the beam, will experience both an attractive force towards the beam and a perpendicular transverse force parallel to the beam surface. Therefore, the vortex will gradually spiral into the beam, finally being trapped, as discussed by Griffiths⁷, and as shown in the cartoon of Fig. 2.

The observed process of vortex capture by the beam is indeed a much more gradual process than the release. In addition to the ‘‘completed’’ events shown in Fig. 2, we also observe many embryonic cases which never fully develop and rapidly revert to the default state, implying that the vortex concerned does not achieve the fully-trapping state, either through some failure of the process, or by premature dislodgement by reconnection with a second vortex.

3. *The authors’ discussion for the mechanism of the frequency shift is particularly unconvincing. They first dismiss the Magnus force based on the experimental observation that the frequency change is nearly independent of the beam’s velocity. However, that is what should happen: the Magnus force has magnitude proportional to velocity, but so does the momentum of the beam. The ratio of these two, which is velocity-independent, determines the frequency shift.*

The supplemental material gives a different argument against the Magnus force, that its direction changes sign during each cycle. On general principle, a force that varies in sync with an oscillator is exactly what would be expected to affect the oscillation. It seems that the authors assume that the frequency shift originates in a change in the average position of the nanobeam, rather than considering the possibility that adding a vertical component to the mainly horizontal motion of the nanobeam might also shift the frequency of the mode.

We have to thank the referee for this comment, which forced us into a useful rethink of the Magnus effect. All we can do is put our hands up here and say that the treatment in the paper of this effect was less than optimal.

In response to the transverse velocity of the beam, the Magnus process applies an oscillating vertical force on the beam at the transverse resonant frequency. No net time-average force is applied, so there is no resultant static deformation of the beam. Furthermore, since the beam is a very high-Q resonating system, only an oscillating force precisely at the vertical resonance frequency will excite the beam into vertical motion. Consequently, this force causes no displacement of the beam, and thus plays no role in increasing the beam tension, and thus the frequency, when a vortex is trapped. We should have got this right first time.

We have amended the text in the paper as below (removing all reference to the Magnus force, which is now irrelevant), and have dealt similarly with the supplemental material.

The identification of the capture of a singly-quantized vortex by the beam is confirmed by several observations. First, the captive vortex generates an additional restoring force increasing the beam's resonance frequency. This arises from the attractive interaction between the trapped vortex and its image in the nearby substrate. The interaction of the vortex with the parallel image vortex gives rise to a static force $F = \rho v_s \times \kappa$, with ρ the fluid density, v_s the superflow created by the image at the position of the beam's vortex and κ the circulation⁶, which displaces the beam towards the substrate, thereby increasing its tension and thus yielding an increased beam resonant frequency when a vortex is trapped along it. For a fuller treatment see the Supplementary Information.

4. *The authors suggest another mechanism, that the interaction of a trapped vortex with the nearby surface creates a steady vertical force that changes the equilibrium location of the nanobeam and hence its tension. They refer to the force as “repulsive,” but in fact it is attractive. (The image vortex is directed opposite to the physical vortex.) Perhaps the authors caught this error while editing; they do say the beam will “sag,” which is consistent with an attractive force. The expected frequency shift from this mechanism is a few times smaller than what is observed.*

Sorry, that is muddle remaining from the writing. For a constrained vortex the force is certainly attractive, which we have amended in the text.

We have amended the text to make this clearer, see response to comment (3) above.

5. *For the trapping of vortices from the nanobeam, there seems to be a gap between the velocity where the tuning fork first produces turbulence and that at which vortices appear on the nanobeam. Do the authors have any explanation?*

Yes, indeed. From earlier studies^a, we find that there is indeed a velocity gap between the initial growth of remnant vortex attached to the tuning fork, indicated by the kink in Fig. 3a, and the actual release of quantum vortices into the bulk fluid, seen as the detection of events by the nanobeam in the same figure.

^aFor a more detailed analysis of this effect on a tuning fork alone, see PRB **94**, 214503 (2016).

We have added a note to this effect in the text:

.... We only detect vortices at tangle densities corresponding to fork velocities above 73mms^{-1} . At this velocity, the rate of detection is very low with the shortest waiting time between the events being $\sim 40\text{s}$ and the longest $\sim 1000\text{s}$. (The gap in the fork driving force between the onset of vortex generation and actual detection by the beam, arises because at fork velocities just above onset, only a dilute gas of individual microscopic vortex rings is emitted. Only at higher velocities does the generated vortex ring density become high enough to favour the ring-ring collisions and reconnections necessary for the production of developed turbulence, see Ref.¹⁷).

6. *I also wonder about the schematic in Figure 2, with a vortex terminating midway along the beam. Is this physically reasonable? Where would such a vortex come from, since a finite segment certainly cannot terminate in the bulk fluid?*

As mentioned above, we initially believed that it is difficult for a section of free vortex approaching the beam to become trapped because of the transverse motion generated by the image vortex. We now know this is not the case, so the original proposal presented in the figure is no longer relevant. This is discussed in the answer to comment (4) above. The text has been amended to make this possibility clearer. However, that said, we do appreciate that a vortex pinned near the beam must have its other end pinned somewhere else on the surface of the container.

7. *The authors note that the timing for the process of freeing a vortex strangely requires the presence of vortices in the fluid but does not depend on their density. However, from the lifetime of the vortex on the nanobeam they then calculate a typical vortex separation. How can this be meaningful if they know that changing the vortex separation does not alter the distribution of lifetimes? I also question the calculation. Why is 3 ms used as the lower end of the trapped lifetime? There are clearly many events in Figure 4 with lifetime 1 ms or even shorter. Secondly, the lifetime and separation ranges seem inconsistent. Each extends about a factor of 3, but the square-root relationship suggests that a factor of 3 in distance should correspond to a factor of 10 in lifetime.*

The interpretation of Figure 4 shows assumes that the vortex trapped on the beam interacts with closest vortex line of the correct orientation and provides an overall idea of how far away vortices are from the beam. On average the distance to vortex varied very little despite the large time span. The time range changes a factor of 30 (3 to 100 ms), while the distances are factor 3 different. There is also a misprint in the original which should have read 40 μm to 230 μm , not 70 μm . We have extended the range to (1 to 100 ms) and distance range is 23 μm to 230 μm .

8. *Can the probability distribution of Figure 4 further test ideas on trapping and release? Is the square root relation between lifetime and distance consistent with Figure 4? Or what about the idea that many incipient vortices are dislodged by the same mechanism that eventually frees the fully trapped vortices? The lifetime curves of Figure 4 overlay, but the curves of Figure 3b showing the time to trap do not. Doesn't this mean that the mechanism corresponding to Figure 4 cannot also be the main factor in Figure 3b?*

This is exactly the point we are trying to make, that for same tangle trapping and release are not the same process. The beam itself does not produce turbulence, its velocity is too slow so there are no incipient vortices or vortex mill. It purely acts as a detector. The great differences between the processes is emphasized in the text but we have also accentuated this in the figure captions of Figs 3 and 4, see answer to (2) above.

9. *I also have two minor questions for the authors. What are the black circles in Figure 3a? The other colours correspond to curves labelled in Figure 3b.*

The black circles in Fig. 3a represent other recorded datasets that have not been included in later graphs. This has been made clear in the caption:

FIG. 3. (Colour online) b. A probability density function of the wait time between events $t_{\gamma\alpha}$ at the same fork velocities. The solid lines correspond to exponential fits, of the form $\propto \exp(-t/\tau)$. Note symbol colour matching between panels a and b. Black points in panel a represent data from other runs not used for panel b. For details, see text.

10. *Also, how are the PDFs of Figure 4 constructed? I do not understand why the lifetime values at small times do not agree better across the data sets, especially for the 75 mm s^{-1} and 90 mm s^{-1} sets which judging from the long-time region have an almost identical number of events.*

The quality of the data at small times is limited by our experimental time resolution. Most of the data come from the heat map style presentation as in Fig. 2 but for small times we have to strain things with single frequency methods. Given the spread of the data, the data sets for each fork velocity are consistent with the same PDF vs lifetime dependence. As said above, that is a major puzzle. Possibly the quite complex geometry of the beam construction means that by chance there are one or more pinning points which attract vortices to the locality, independent of the bulk vortex density.

REVIEWER COMMENTS

Reviewer #1 (Remarks to the Author):

The manuscript has significantly improved after the revision. I can recommend publication in Nature Communications. I have only minor comments:

1. On page 3: "Therefore in all our measurements the beam response is linear and virtually dissipationless." Actually, Q-value is not given explicitly and origin of the dissipation is not directly discussed. (Only in passing in Sec. IIIc of Supplementary material).

2. There is still some confusion with the "local vortex density". One reference to it remained on page 7 and in the next paragraph the authors suggest that the beam is sensitive to "immediate surrounding of the beam, say within a distance of a few micrometres." Simultaneously on page 9 the authors say that expected intervortex distance in the tangle is $\sim 100\mu\text{m}$. How can vortex density be defined on a scale of a "few micrometres" when intervortex distance is of the order of 100 micrometers?

Reviewer #2 (Remarks to the Author):

The authors answer the questions properly and made the required revisions on the manuscript. I could recommend the acceptance of this paper.

Reviewer #3 (Remarks to the Author):

I thank the authors for their thoughtful responses. The manuscript reads well and certainly merits publication somewhere. I remain less sure that it is a good fit for a general-audience journal like Nature. They present a new technique that applies only to a very specialized system, and they show convincingly that different mechanisms govern the trapping and releasing of vortices from their

beam. They are hopeful that their nanobeams will lead to useful information on quantum turbulence, but the path to get there is not yet clear.

I agree with the first referee on the importance of how the beam affects the surrounding flow, and on the limitations of the method if the interaction with a wall is necessary. The claim in the final sentence of the abstract that the beams are "capable of probing turbulence on the micron scale" does not yet seem justified. The concluding paragraph of the manuscript is a more realistic assessment of what has been accomplished and the likely next steps.

I appreciate the authors' rethinking of the Magnus force, but it still seems to me that the Magnus force may be a factor in their observations. Even if the vertical and horizontal resonant frequencies of the beam differ substantially, the addition of a Magnus force (by virtue of having a vortex attached to the beam) couples these vibrational modes. The originally horizontal oscillation may become an ellipsoidal motion with a horizontal major axis. There will also be some frequency shift. Depending on the difference between the original horizontal and vertical frequencies, that shift could easily be in the few kHz range. One problem with the authors' idea that the frequency change comes from the beam's attraction to the nearby surface is the marginal agreement with observation. In the appendix the authors calculate a 1 kHz shift, compared to 3 kHz in Figure 2. Furthermore, that 1 kHz is based on the maximum extra tension, which occurs only at the midpoint of the beam, so the discrepancy would probably increase in a more thorough calculation. The authors can test this in their subsequent work by varying the distance between the beam and the wall. This should have a strong effect with their proposed mechanism but very little if the Magnus force is responsible for the frequency shift.

Comments to reviewer 1

We are pleased that this reviewer is happy with the paper, *The manuscript has significantly improved after the revision. I can recommend publication in Nature Communications ...* apart from minor comments, which we can readily deal with, as below:

1. *On page 3: “Therefore in all our measurements the beam response is linear and virtually dissipationless.” Actually, Q -value is not given explicitly and origin of the dissipation is not directly discussed. (Only in passing in Sec. IIIc of Supplementary material).*

That is a fair point, we have added the following passage on the dissipation mechanisms and the Q -values in the manuscript:

The beam has a vacuum frequency of 2.166 MHz, and at the measurement field of 5 T has a $Q \sim 2.8 \times 10^3$. The beam is driven at a velocity of only a few millimetres per second. This is orders of magnitude below the expected velocity for the onset of turbulence production¹¹. Therefore, in all our measurements the beam response is linear. The frequency width arises almost entirely from intrinsic losses in the beam material. While on operating the beam in the liquid we do see a vanishingly small increase in the damping above the vacuum value (from a very small level of acoustic emission into the liquid), the contribution from the superfluid is essentially negligible.

2. *There is still some confusion with the “local vortex density”. One reference to it remained on page 7 and in the next paragraph the authors suggest that the beam is sensitive to “immediate surrounding of the beam, say within a distance of a few micrometres.” Simultaneously on page 9 the authors say that expected intervortex distance in the tangle is $\sim 100 \mu\text{m}$. How can vortex density be defined on a scale of a “few micrometres” when intervortex distance is of the order of 100 micrometers?*

The referee comment here holds for the instantaneous situation. However, that is not quite the case here. We are measuring a time average over long periods. Even if the intervortex distance is $100 \mu\text{m}$, vortices will still penetrate into the region of a few micrometers of the beam, can be detected, and an estimate of the intervortex spacing, even if much larger, can be made from those interactions.

Hopefully this will put the referee’s reservations to rest.

Comments to reviewer 2

The authors answer the questions properly and made the required revisions on the manuscript. I could recommend the acceptance of this paper.

We are certainly happy with that.

Comments to reviewer 3

We are thankful that our paper raised the discussion. *I thank the authors for their thoughtful responses. The manuscript reads well and certainly merits publication somewhere. I remain less sure that it is a good fit for a general-audience journal like Nature. They present a new technique that applies only to a very specialized system, and they show convincingly that different mechanisms govern the trapping and releasing of vortices from their beam. They are hopeful that their nanobeams will lead to useful information on quantum turbulence, but the path to get there is not yet clear.*

We would seriously disagree that this is of “*niche*” interest. Quantum turbulence may be characterised as a very specialized system. However, it is one which has the potential to throw light on the problems of classical turbulence, and is a field which has rapidly expanded recently. As to the subject not being a good fit for a general Nature audience, the Nature stable of journals along with Science have published around 20 papers on this subject in recent years (including a number of ours).

Furthermore, on the usefulness of these devices, “*the path to get there*” is indeed becoming very clear. We should not be saying this here, but the situation on the ground is moving on and we are now able to use these beams to detect the energy bursts emanating from nearby vortex reconnections and are also using them as attachment points where we can pin the end of a “*free*” vortex half way along the beam and use the beam to oscillate the pinned end to generate Kelvin waves along it. (Not for the public domain, please.) These beams are actually proving to be a serious boon to our work.

I agree with the first referee on the importance of how the beam affects the surrounding flow, and on the limitations of the method if the interaction with a wall is necessary. The claim in the final sentence of the abstract that the beams are “‘capable of probing turbulence on the micron scale’” does not yet seem justified. The concluding paragraph of the manuscript is a more realistic assessment of what has been accomplished and the likely next steps.

In future we can certainly create beams without a backing plane which would put them in a “bulk” liquid situation. However, that said, we are certainly probing the nearby vorticity and that is certainly on the micron scale. The propinquity of the ground plane may influence that to some degree but we have to start somewhere.

I appreciate the authors’ rethinking of the Magnus force, but it still seems to me that the Magnus force may be a factor in their observations. Even if the vertical and horizontal resonant frequencies of the beam differ substantially, the addition of a Magnus force (by virtue of having a vortex attached to the beam) couples these vibrational modes. The originally horizontal oscillation may become an ellipsoidal motion with a horizontal major axis. There will also be some frequency shift. Depending on the difference between the original horizontal and vertical frequencies, that shift could easily be in the few kHz range. One problem with the authors’ idea that the frequency change comes from the beam’s attraction to the nearby surface is the marginal agreement with observation. In the appendix the authors calculate a 1 kHz shift, compared to 3 kHz in Figure 2. Furthermore, that 1 kHz is based on the maximum extra tension, which occurs only at the midpoint of the beam, so the discrepancy would probably increase in a more thorough calculation. The authors can test this in their subsequent work by varying the distance between the beam and the wall. This should have a strong effect with their proposed mechanism but very little if the Magnus force is responsible for the frequency shift.

In principle, a trapped vortex could indeed couple the vertical and horizontal modes of the beam creating an ellipsoidal motion and thus introducing a Magnus force. However, when we put the numbers in this is of vanishing probability. Since the beam is very far from having a perfect square cross-section (in fact, with its measured 130×200 nm profile of almost a factor of two difference), the horizontal and vertical frequencies are likely to differ by a frequency of order a MHz. Given that the vertical mode will also be very narrow with Q-value also in the 10^3 range, we believe that any overlap which could possibly couple the modes will be utterly negligible, and that any influence of the Magnus force can safely be neglected.

In defence of our original estimate of the beam's attraction to the surface, the geometry of the system is so complex, our incomplete knowledge of the beam tension (needing to be pre-stressed at room temperature) and the precise dimensions of the beam, mean that the uncertainty in our estimated frequency is certainly consistent with the measured result. That said, there are probably a number of further factors which might influence the trapped/untrapped frequency difference, but in the end we do indeed see a frequency shift which does indicate that vortices are indeed trapped and can use that information.